# Decline in Honeybees and Its Consequences for Beekeepers and Crop Pollination in Western Nepal

**DOI:** 10.3390/insects15040281

**Published:** 2024-04-16

**Authors:** Susanne Kortsch, Thomas P. Timberlake, Alyssa R. Cirtwill, Sujan Sapkota, Manish Rokoya, Kedar Devkota, Tomas Roslin, Jane Memmott, Naomi Saville

**Affiliations:** 1Spatial Foodweb Ecology Group, Organismal and Evolutionary Biology Research Programme, Faculty of Biological and Environmental Sciences, University of Helsinki, Viikinkaari 1, 00014 Helsinki, Finland; alyssa.cirtwill@gmail.com (A.R.C.); tomas.roslin@helsinki.fi (T.R.); 2Tvärminne Zoological Station, Faculty of Biological and Environmental Sciences, University of Helsinki, J.A. Palménin tie 260, 10900 Hanko, Finland; 3School of Biological Sciences, University of Bristol, Bristol Life Sciences Building, 24 Tyndall Avenue, Bristol BS8 1TQ, UK; thomas.timberlake@bristol.ac.uk (T.P.T.); jane.memmott@bristol.ac.uk (J.M.); 4HERD International, Thapathali, Kathmandu 24144, Nepal; sujan.sapkota@herdint.com; 5Nepal School of Public Health, Karnali Academy of Health Sciences, Jumla 21200, Nepal; manishrokaya449@gmail.com; 6Faculty of Agriculture, Agriculture and Forestry University, Rampur, Chitwan 44200, Nepal; kdevkota@afu.edu.np; 7Department of Ecology, Swedish University of Agricultural Sciences, Ulls väg 18B, 75651 Uppsala, Sweden; 8Institute for Global Health, University College London, 30 Guilford Street, London WC1N 1EH, UK; n.saville@ucl.ac.uk

**Keywords:** *Apis cerana cerana*, beekeeping, climate change, crop production, insect decline, insect pollination, pollinator decline, smallholder farming

## Abstract

**Simple Summary:**

Beekeeping provides numerous economic, cultural, and crop pollination benefits to communities around the world, including in Nepal, where traditional beekeeping has been practiced for centuries. Anecdotal reports suggest that the native Asian honeybee *Apis cerana* is declining across Asia, but few studies have measured the extent of these declines or their implications for beekeepers and farmers. Working in the Jumla District of Western Nepal, our aims were to investigate population trends of the native honeybee *Apis cerana cerana* and assess its importance for livelihoods and crop pollination. Interviews with 116 local beekeepers revealed a 44% decline in the number of occupied beehives and a 50% decline in honey production per hive from the years 2012 to 2022. Beekeepers reported climatic changes and the loss of flowers as the main drivers of this decline. These declines pose a major threat to local communities, as sales of honey contribute 16% of total household income for beekeepers, and *Apis cerana cerana* is a key pollinator of many important crops. Our study provides a warning signal of potential declines in all insect pollinators across the region and calls for actions by farmers, beekeepers, researchers and policy-makers to work together in addressing this socio-ecological crisis.

**Abstract:**

In understudied regions of the world, beekeeper records can provide valuable insights into changes in pollinator population trends. We conducted a questionnaire survey of 116 beekeepers in a mountainous area of Western Nepal, where the native honeybee *Apis cerana cerana* is kept as a managed bee. We complemented the survey with field data on insect–crop visitation, a household income survey, and an interview with a local lead beekeeper. In total, 76% of beekeepers reported declines in honeybees, while 86% and 78% reported declines in honey yield and number of beehives, respectively. Honey yield per hive fell by 50% between 2012 and 2022, whilst the number of occupied hives decreased by 44%. Beekeepers ranked climate change and declining flower abundance as the most important drivers of the decline. This raises concern for the future food and economic security of this region, where honey sales contribute to 16% of total household income, and where *Apis cerana cerana* plays a major role in crop pollination, contributing more than 50% of all flower visits to apple, cucumber, and pumpkin. To mitigate further declines, we promote native habitat and wildflower preservation, and using well-insulated log hives to buffer bees against the increasingly extreme temperature fluctuations.

## 1. Introduction

Insect pollination enhances the yields and quality of 75% of global crop plants [1], but pollination services are severely threatened globally [2]. Declines in insect pollinators have been reported across most regions of the world where historic data and monitoring programs enable such assessments [3,4,5,6]. These declines are thought to be driven by a combination of anthropogenic factors including agricultural intensification, habitat loss, disease, pesticide use and climate change [7,8,9,10]. The threats facing pollinators likely differ amongst regions of the world and the impact of pollinator declines on food security is potentially the most severe for smallholder farmers in low-income countries. These farmers strongly rely on local pollination services for their food and nutrition security, as well as their income [11,12]. Despite the immense importance of pollination services to smallholder farmers, we know almost nothing about the population trends of insect pollinators in regions of the world where smallholder farming systems predominate [13]. This makes it challenging to assess the degree of risk and identify likely drivers and mitigation strategies.

The lack of historic data and formal monitoring schemes for insect populations in most low-income countries currently poses a major barrier to understanding global pollinator statuses and trends and the ability to tailor conservation efforts accordingly. In the absence of long-term quantitative data on insect population trends in understudied regions, other forms of data may prove useful, including the records and perceptions of local beekeepers. Given the strong reliance on bees for their livelihoods, food, and medicinal needs, beekeepers typically have a strong awareness of the health and well-being of their bees and the environment in which they forage [14]. This should make beekeepers ideal witnesses for reporting on the population status of the bees they keep, along with the environmental stressors affecting them. However, despite their wealth of local knowledge, the perceptions, records and experience of indigenous beekeepers are rarely considered [2].

Beekeeping is practiced widely across the world, predominantly using two species of managed honeybees, *Apis mellifera* Linnaeus (1758) and *Apis cerana* Fabricius (1793), but also various stingless bees such as *Trigona* spp. and *Melipona* spp. [15]. Beekeeping has immense cultural and economic importance across the world [16], particularly for smallholder farmers, as it requires only a small amount of land and minimal investment whilst providing a significant source of income [17]. In addition to producing valuable products such as honey, beeswax, pollen, and propolis, managed bees can play a crucial role in crop pollination, substantially enhancing agricultural yields [18,19]. In Nepal, where this study is based, traditional beekeeping of the native *Apis cerana* has been practiced for many centuries. Bees were traditionally kept in wooden log hives, but commercial beekeeping with the rearing of *A. cerana* in modern beehives began in the 1980s [20]. After the introduction of *Apis mellifera* in 1992, commercial and migratory beekeeping was taken up in the plains and mid-hills of Nepal [18].

The introduction of *A. mellifera* in Nepal reflects a broader trend of the introduction and replacement of *A. cerana* across South Asia, resulting in widespread reports of *A. cerana* population decline [21]. Here, we focus on *Apis cerana cerana* Fabricius (1793), a subspecies of *A. cerana* native to the western high hills and mountains of Nepal, where it occurs both as a wild and semi-domesticated pollinator. Although the honey production of *Apis cerana cerana* is relatively low compared to the western honeybee *Apis mellifera*, *Apis cerana cerana* is an efficient pollinator in harsh, high-altitude mountain environments such as in the Jumla District of Western Nepal [22]. In the Jumla District, *Apis cerana cerana* was identified as the most important pollinator in both subsistence and commercial cash crop farming systems [23], consistent with the high importance of *Apis* spp. honeybees worldwide [1,24]. Moreover, *Apis cerana cerana* requires relatively low maintenance and management costs, and beekeepers possess indigenous technical knowledge on their forage and management [25], making the species highly suitable for remote subsistence-farming communities with limited access to funds and technology [17,22].

The main objective of this study is to shed light on the management practices, population trends and conservation status of *Apis cerana cerana* for Jumla District—a remote mountainous region of Western Nepal and understudied region of the world. The population status of *Apis cerana cerana* is unclear in Nepal since local-level population assessments are lacking. Moreover, the implications of potential changes in bee pollinators for local crop production and beekeeper livelihoods are unknown and have not been previously investigated. Here, we address the ecological and economic importance of *Apis cerana cerana* for beekepers in Jumla, using a combination of different approaches. We asked the three following questions: (1) Have beekeepers experienced a change in the population size of *Apis cerana cerana*? (2) Has the honey yield per beehive and the number of beehives per beekeeper changed between the years 2012 and 2022? (3) What are the drivers of *Apis cerana cerana* population change, and what consequences might this have for crop production and beekeeper livelihoods?

## 2. Materials and Methods

### 2.1. Study Site and Sampling Methods

The main beekeeper study was conducted in 10 smallholder farming villages at 2400–3000 m above sea level in Patarasi Rural Municipality, Jumla District, Mid-Western Nepal (Appendix A, Figure A1 and Figure A2 and Table A1). Jumla is a remote mountainous district, situated in the Karnali Province of Nepal. Rates of poverty, food insecurity and malnutrition are particularly high in this region, and 80% of the population is directly dependent on smallholder agriculture [26]. More than 50 crops are grown in this region, including many pollinator-dependent species such as apples, beans, pumpkins, cucumber, mustard, and buckwheat. Beekeeping with traditional log hives has been practiced in Jumla for centuries, with many households keeping hives of the native *Apis cerana cerana* in and around their homesteads [25]. Despite efforts to test and introduce improved, modern beehives in the 1990s [25], most beekeepers keep their bees in hollowed-out log hives, which prevent combs from being inspected without permanently damaging them. *Apis cerana cerana* is the only bee subspecies that is widely kept in this region, as it is higher-yielding and less likely to abscond than other subspecies such as *Apis cerana himalaya* and *Apis cerana indica*. To avoid the spread of disease, the non-native *Apis mellifera* is not currently kept in this region. The honey produced by *Apis cerana cerana* hives is either sold for income or retained for household consumption, medicine, and use in religious ceremonies. Since honey production is one of the few sources of cash income for local farmers in the region, beekeeping is an important activity for sustainable livelihoods. Beehives are generally situated near houses, surrounded by small vegetable gardens and livestock paddocks. Village areas also include small arable fields, apple orchards and large areas of steep, heavily grazed grassland pasture and native coniferous forest (Appendix A, Figure A2).

### 2.2. Questionnaire Survey and Interview

We conducted questionnaire surveys on a total of 116 beekeepers in the 10 study villages to explore the status of honeybee populations (*Apis cerana cerana*) in Jumla. On average, 11 beekeepers were interviewed per village, which represented the majority of the core beekeeping population in the smaller villages and about half for the bigger villages (see Appendix A, Table A1, for the exact number of beekeepers per village). The questionnaire was designed to reveal temporal changes in honey yield per hive and number of (occupied) hives per beekeeper currently (i.e., in the year 2022) and in the past (years: 2021, 2019, 2017, 2012, and before 2012). To better understand the potential drivers behind any changes, we asked the beekeepers to rank the main causes for changes in number of honeybees, beehives and honey yield. Similarly, to gain insights into the potential consequences for their livelihood, we asked the beekeepers to rank the main impacts of honeybee declines on their livelihood. The questions and reply options are shown in Appendix B. Interviews were conducted in the Nepali language by a trained data collector in the presence of the lead author and answers were recorded on Android tablets using a custom-built data collection app in CommCare Version 2.48.3 (http://www.commcarehq.org/home/; accessed on 30 August 2022), a cloud-based data collection platform. All interviews were performed by the same individual to avoid any variation due to interviewer effects.

To test for temporal trends in honey yield per hive and number of beehives per beekeeper, we fit two generalised linear models (GLMs) relating honey yield (in kg) and number of occupied hives to year. Because of right-sided skewness in the data, we used a log link function in both cases. Honey yield was modeled using an identity (Gaussian) link function and the number of beehives using a log (Poisson) link function. One was added to the observed values to account for zeros. Predictions were back-transformed to the original scale for easier interpretation. Assuming that the log-linear relationships may continue in the future, we predicted future changes in honey yields and number of beehives up to the year 2030. The regressions were performed using base R, version 4.3.0 [27], and data visualizations using ggplot [28].

To gain a more general and in-depth understanding of the status of honeybee populations in the Jumla District and the drivers of population change, we conducted an in-depth interview with one carefully selected local, lead beekeeper who comes from a household which has kept large numbers of traditional beehives for many generations. The lead beekeeper worked in the Jumla veterinary office for many years and is also one of the few people who uses the improved Jumla top-bar hive to manage bee colonies. Because of this, he has better technical knowledge of beekeeping than most beekeepers in the district (see Appendix C for a transcription and translation of the interview from Nepali to English). In the interview, the same questions were asked as in the questionnaire survey, but the lead beekeeper was encouraged to elaborate on his replies.

### 2.3. Economic Importance of Honeybees in Jumla District

To assess the prevalence of beekeeping and its importance as a livelihood strategy across the wider Jumla District, we conducted a series of brief structured questionnaires with the lead farmer of 920 households across all eight municipalities of Jumla District (Chandannath, Kankasundari, Sinja, Hima, Tila, Guthichaur, Tatopani and Patarasi). Approximately four respondents were randomly selected from each village in each of the eight municipalities during a series of farmer consultation meetings. As far as possible, the gender ratio of the respondents was balanced to ensure roughly equal participation in the study. Each respondent was asked a simple binary question of whether or not they kept bees. If they answered yes to this question, the participant was then asked how many occupied beehives they had and how much income they derived in the previous year from selling honey. Finally, the respondent was asked to list all of the crops they grow and report their total annual household income from the sale of all agricultural produce. These data enabled us to assess the prevalence of beekeeping in the region and to calculate the proportional contribution of honey sales to total agricultural revenue and determine the relative value of beekeeping as a livelihood strategy. These wider household surveys were conducted by eight trained data collectors (one in each municipality) and answers were recorded in the open-source data collection platform Open Data Kit (ODK; (https://getodk.org/; accessed on 12 Novenber 2022).

Written informed consent was obtained from all participants in the beekeeper and household surveys, and from the lead beekeeper, to publish the information obtained through the interviews conducted in this study.

### 2.4. Crop Dependency on Apis cerana cerana

To investigate the Asian honeybee’s importance as a crop pollinator in Jumla, we analyzed an insect visitation dataset collected in the 10 villages in Jumla during the same time period as the beekeeper survey was conducted. Insect visitation sampling was conducted every two weeks from 18 April to 4 November 2021 (spring to autumn) in a 600 × 600 m sampling area centered on the midpoint of each study village. This area was divided into three habitat categories: village, crop, and semi-natural vegetation (Appendix A, Figure A2). In each of these habitats, we randomly located three replicate fixed survey plots of 60 × 60 m (9 plots per village). Every two weeks, a 40 min. plant-pollinator visitation survey was conducted in each plot to record the interactions between plants (both crop and non-crop species) and flower-visiting insects. Insects were captured, pinned and identified to species or morphospecies (see acknowledgements). For full details on the insect visitation sampling, see Section D.1 and [23].

For each crop in each village, we calculated the proportion of all visits to each crop that were made by *Apis cerana cerana*. Restricting our analysis to pollinator-dependent crops only, we ranked crops from the highest to the lowest proportion of visits made by honeybees, thereby identifying crops which are most reliant on honeybees and therefore most likely to be impacted by honeybee declines.

As well as making a large number of visits to crops, honeybees might be especially important pollinators because they carry large amounts of pollen. To test this, we calculated the pollen carrying capacity of *Apis cerana* and compared it with that of other insect taxa recorded visiting crop flowers in this study region. Pollen carrying capacity was recorded by swabbing a total of 1928 insect specimens (representing 136 unique crop flower-visiting taxa) with glycerine jelly and counting the total number of pollen grains on each insect using light microscopy; pollen in the corbicula was not included in the sampling as this is not available for pollination. For each of the 136 insect taxa that were sampled, we calculated a mean pollen carrying capacity value across all replicate specimens and compared these values amongst taxa to identify the best transporters of pollen (see Section D.2 for further details).

## 3. Results

The district-wide household survey of 920 farming households across all eight municipalities of Jumla revealed that 12.5% of all farming households keep at least one beehive. For those households which kept bees, the mean number of hives per household was 3.4 (±0.3 SE). The average number of beehives for the 116 selected beekeeper respondents of this study decreased from 10.5 in 2012 to 5.3 in 2022.

### 3.1. Temporal Changes in Honey Yield and Beehives

Across all villages, beekeepers report a significant (*p* < 0.0001) decline in honey yield from (before) 2012 to 2021 (Figure 1a). Specifically, the honey yield declined by 55% over this 10-year period (see Table 1 for glm coefficients). Declines were consistent across all 10 villages (Figure 2), but were only statistically significant in two villages marked with an asterisk in Figure 2.

While we record a few instances of very high honey yields (>15 kg per hive), these are likely to represent recall errors by the respondents rather than true values, possibly because respondents provided a high guess instead of stating that they did not recall the honey yields for a given year. However, as these high values tend to occur in the more recent years (in 2017 or later), they are unlikely to have influenced the overall trend in the detected honey decline.

Across villages, beekeepers also reported a significant (*p* < 0.0001) decline in the number of (occupied) beehives per beekeeper during the period from (before) 2012 to 2022 (Figure 1b), corresponding to a total decline of 44% during this 10-year period (Table 1). The decline in beehives per beekeeper varied between villages and was statistically significant in Chaura, Chuma, Lorpa, Patmara, and Urthu (Figure 3). The significant decline in beehives in Patmara is especially notable, as this village used to be a hub for beekeeping in the Jumla District in the 1990s and beekeepers there had a rich knowledge of *Apis cerana cerana* and traditional ways to manage them [25].

### 3.2. Reasons for the Decline in Honeybees

According to the beekeepers’ responses to the questionnaire, the four main causes for the decline in honeybees in the Jumla District are (1) climate change (listed as a major driver by 59% of respondents), (2) reduced availability of flowers (55% of respondents), (3) insecticide/herbicide use (48% of respondents), and (4) bee diseases (37% of respondents) (Figure 4). When asked in more detail about the characteristics of the changes in the climate, the beekeepers reported (1) heavier monsoon, (2) more unpredictable weather patterns, and (3) drier weather (see Appendix E, Figure A3). While heavier monsoon (rain) and drier weather may initially seem contradictory, the issue here is shifts in the timing of these events, which may influence the timing of flowering plants and insect activity, potentially causing mismatches between the two. When asked in more detail about the characteristics of lower flower availability, the beekeepers reported (1) cutting of the forest, (2) fewer flowering crops, (3) overgrazing, and (4) wetter weather as the main causes for the decline in flower abundances (see Appendix E, Figure A4). Some respondents also mentioned the destruction of the deciduous flowering shrub known as ‘Dhatelo’ (*Prinsepia utilis*) as a significant factor in reducing flower availability. ‘Dhatelo’ shrubs bloom early in the season when few other plants are in flower. Thus, their removal is likely to increase food scarcity for honeybees early in the season with negative impacts. The decline in Dhatalo is complex and was attributed to a number of reasons by the respondents. Traditionally, Dhatalo was used as a fence in crop fields, especially in fields growing leguminous plants often found on sloping areas near forests, to protect them from domestic and wild animals. However, the accessibility of modern iron fences have made them a more convenient alternative to Dhatalo (which is a thorny plant) and therefore the plants have been removed from the fields. The general conversion of semi-natural lands to farming land might also have been destroying Dhatelo. Moreover, the need of Dhatalo for household purposes (Dhatelo seed oil) has decreased due to the availability of commercial alternatives such as sunflower and soybean oil, yet another reason why the plants have been removed.

### 3.3. Importance of Apis cerana to Agriculture

Out of all the pollinator-dependent crops grown in this study region, apple *Malus domestica* is the most dependent on honeybees (67% of all visits to flowers of this crop were made by *A. cerana cerana*) followed by cucumber *Cucumis sativus* (52%), pumpkin *Cucurbita maxima* (50%), mustard seed *Brassica alba* (42%), and chilli *Capsicum* sp. (33%). These crops are all important to local farmers and are widely grown in the region. Particularly important are chilli, apple, and pumpkin, which are grown by more than 80% of all farmers (Figure 5).

In addition to the high number of visits made by honeybees to important crops, the bees also carry a relatively large amount of pollen per individual bee. This means they can potentially deposit many pollen grains per visit, which can be important for full pollination of fruits such as apples. Honeybees carried, on average, 96 (±6.8 SE) pollen grains per individual, compared to a mean of 57 (±7.7 SE) and a median of 18 grains across all taxa sampled. Although there were 29 other insect taxa (out of a total of 135) which carried more pollen than honeybees (mostly bumblebee and solitary bee species), honeybees were nevertheless within the top 20% of pollen-carrying taxa (see Appendix D for how the pollen data were collected). The high visitation rates of honeybees to a range of important crops, combined with their relatively high pollen transport capacity, suggests that *Apis cerana cerana* is amongst the most important crop pollinators in this region. The relatively low number of visits from honeybees to buckwheat in our study (7% of all buckwheat visits were made by honeybees) may be surprising, as honeybees are often dominant visitors to buckwheat at lower elevations [29]. However, the buckwheat fields in our study region are typically located on the hills above villages and therefore at relatively high elevations (>2500 m), where flies are more dominant than honeybees [11].

Apples are the most economically important crop in this region, and their high dependence on honeybees suggests their production may be heavily impacted by the observed honeybee declines. Consistent with this prediction, when asked about changes in apple yield quality and quantity, 55% and 42% of the beekeepers responded that they experienced a decrease in apple yield and change in quality, respectively, whereas only 8.6% and 2.6% experienced an increase, respectively (see Appendix E, Figure A5).

### 3.4. Effects on Beekeepers’ Livelihoods

Half of the beekeepers (51%) reported impacts to their livelihood due to loss of honeybees, 42% reported no change and 7% did not know (see Appendix E, Figure A6). For the beekeepers that experienced a negative impact on their livelihood, the main features were loss of income and less, or no, honey for their own consumption. In a follow-up question, the 42% of the respondents who reported no impact replied that they have other income from high-value crops such as apples (see Appendix E, Figure A6).

A wider survey of 920 farming households across all eight municipalities of the Jumla District revealed that the mean annual household income from the sale of honey was NPR (Nepalese rupees) 5493 (±1015 SE), or USD 41.23, representing 16% of total household farming income (mean NPR 34,567, or USD 260). Given the rate of decline in honey yield reported in this study (Figure 1), we estimate that beekeepers have lost approximately 14% of their household income from agriculture over the last ten years as a result of these changes. This ignores any loss of income as a result of crop yield declines, which may be even more substantial. Interestingly, despite the heavy reliance on crop pollination services by farmers in this region, only 17% of respondents in the wider household survey were aware that honeybees play an important role in crop production through their provision of pollination services.

### 3.5. Summary of In-Depth Interview with Key Informant

The key informant beekeeper (local expert) confirmed the decline in honey yield indicated by the beekeepers of the survey. The total amount of honey harvested from all of his hives combined has declined from 50–100 kg (from 25–35 hives) per year to only 3 kg (nine hives) over the past 5 to 10 years. He mentioned that the amount of honey produced was not even sufficient for keeping his bee colonies alive during the winter. He further stated that the bees are starving. The two most important drivers for the decline in honeybees that he mentioned were climate change and pesticide use. In particular, he stated that there is a huge effect of climatic change on bees: “It rains when is should not, and it does not rain when it should”. Similar descriptions were given by several beekeepers during the survey. For example, when bees start to become active in April and May, it rains heavily when it should not be raining. According to the key informant’s perception, the heavy rainfall will wash away the nectar which, combined with fewer flowers, decreases the resource availability for the bees. Moreover, he said that climate change induces shifts in flowering seasons and behavior. Plants flower earlier and the season is shorter. He believed that this is why bees do not have enough food and die. According to him, there is no problem of livestock overgrazing in Jumla. In terms of pollination services, the key informant stated that apple quantity and quality has diminished compared to the past. When asked about how the decline in honeybees has affected his livelihood, he confirmed that the decline in honeybees has affected his livelihood negatively due to loss of income and loss of honey for household consumption. He also said that farmers are not prioritizing beekeeping as much as in the past. According to him, the farmers are unaware of the importance of bees for crop pollination and food production, which may explain why they do not prioritize beekeeping.

## 4. Discussion

According to the beekeepers in our study, the honeybee *Apis cerana cerana* is declining in Western Nepal. The reported honeybee decline is likely indicative of a regional trend, consistent with previous reports of declining *Apis cerana* populations in Nepal and throughout Asia [21,30]. Because of declining bee populations, honey yields and the number of occupied beehives are also decreasing. Changing climate and unpredictable weather patterns and reduced flower availability are the major drivers for the declines as perceived by the beekeepers. Locally, the decline in bees is associated with loss of income from honey sales and poorer crop pollination, especially for apples. This poses a risk to the livelihoods and food security of beekeepers and smallholder farmers in the region.

### 4.1. Are Honeybees Starving in Western Nepal?

Across the world, pollinating insects are threatened by a range of anthropogenic pressures such as agricultural intensification, habitat loss, climate change and pesticide use [7,8,9]. All of these pressures are known to reduce the availability of food (pollen and nectar) for insects [31], a resource which is considered the most important factor limiting pollinator populations [32,33]. Honeybees are generalist foragers whose accumulation of honey is largely determined by the quantity and quality of floral resources present in the landscape, as well as local climatic conditions which influence their ability to forage [34]. The declining honey yields reported by beekeepers in our study region therefore indicate a reduction in floral resource availability or a reduced ability to freely forage and collect these resources, for example due to unfavorable weather conditions. Weather conditions are a key factor influencing the foraging activity of both managed honeybees and wild bees [35,36] and this may be particularly relevant in our study region where weather patterns have changed dramatically as reported by beekeepers and confirmed by empirical data [37,38]. In particular, beekeepers report that heavier and more erratic rainfall events—often coinciding with important stages of the colony life cycle—are impacting the ability of bees to forage and accumulate honey reserves.

Fewer flowers and reduced foraging ability equate to less food for bees and will result in honeybees collecting less pollen and nectar for their brood—thus reducing the size of colonies and limiting the honey stored for the winter, which ultimately risks starvation of the bee colonies. In the case of our study region, where beekeepers report climate change and reduced flower availability as the two most important drivers of bee and honey declines, it is likely that these two factors are interacting. For example, climate change can alter the distribution, quantity and quality of the flowers on which insects depend for food [39]. Moreover, warming can lead to unpredictable seasonal weather patterns which cause phenological mismatches between plants and pollinators [40]. As insects are ectotherms, changes in temperature can also have a direct influence on their physiology and behavior and thus change their foraging range [41]. For bees, the outcome of such direct and indirect effects on behavior and physiology can influence a colony’s ability to store sufficient honey to survive the winter. Our key informant remarked that honey yields are increasingly becoming too low to even keep colonies alive throughout the winter.

As a factor contributing to reductions in honey yields, reduced flower availability may force bees to forage over greater distances. A recent study using an agent-based model showed that flower abundance is a major driver of foraging distances with decreasing floral abundance, leading to larger foraging distances [42]. Evidence also shows that honey bees produce less honey if they need to forage at greater distances [41,43,44]. Moreover, the breeding rate of a colony partly depends upon the quantity of incoming forage [45]. In other words, if incoming foraging decreases, the colonies decline. Larger foraging distances may also put honeybees at greater risk of exposure to unpredictable weather regimes, increasing the chances of being caught in bad weather. Beekeepers, including the lead beekeeper in our survey, repeatedly stated that bees are often caught in weather conditions with heavy rainfall during flights and therefore do not return to their hives. Increasingly unpredictable weather patterns as predicted for the Himalayan region [38] could worsen the situation for honeybees and other wild pollinators.

### 4.2. Honeybee Decline and Its Consequences for Crop Pollination

*Apis cerana cerana* honeybees are one of the most important crop pollinators in this region, both because of the large number of visits they make to key crops and because of the large amount of pollen they carry per individual [23]. In the unlikely event of total honeybee loss, farmers stand to lose approximately 45% of all crop pollen transport and as much as 73% in the case of apple [23]. Red Delicious apples, the major cash crop in the region, are almost entirely pollinator-dependent and may already be showing signs of decline in honeybee pollination, as beekeepers in our survey generally report reductions in apple quality (e.g., shape and size) and quantity.

In addition to cash crops, honeybees are important pollinators for a range of other nutritionally important crop plants [23] such as mustard seeds, pumpkins, and chilies, which may all experience decreased pollination as a result of honeybee declines. In smallholder farming villages where most people depend upon local food production for survival, these declines have the potential to seriously impact their food and nutrition security, as well as their livelihoods. Surprisingly, only 17% of respondents were aware of the value of honeybees or other pollinators to crop production, which is in line with the key informant’s perception. This lack of pollination awareness amongst smallholder farmers may result from their limited scientific education on concepts such as plant reproductive biology and insect ecology. Thus, although their local knowledge of plants and insects is very high, they do not recognize or understand the importance of these insects in transporting microscopic pollen grains between flowers. This limited awareness may also explain why only half of the respondents reported a negative impact of bee declines on their livelihood, as they only consider the loss of income from reductions in honey production and ignore the importance of bees for crop production and hence agricultural income.

### 4.3. Are Wild Insect Pollinators also Declining in Jumla?

It is likely that honeybees are not the only insects declining in Jumla. Many of the mechanisms that are reported to be driving honeybee decline in Jumla, such as fewer flower resources, unpredictable climatic patterns, phenological mismatches, and agrochemical use, are also known to cause declines in other pollinating insects [2,7,9,10]. Honey yields are known to serve as a convenient bio-indicator of floral resource availability and foraging conditions, so consistent declines in honey yields point towards strong floral resource limitation and resulting population declines for all pollinators in this region [34].

Indeed, many wild pollinators are likely to be affected even more severely than honeybees as they do not retain honey reserves to help them through sparse periods of the year and their food supply is not supplemented by beekeepers, as is sometimes the case for honeybees. Moreover, their foraging ranges are shorter than honeybees making it more challenging for them to reach distant resources. This prediction is consistent with studies from other regions of the world which indicate that declines in wild pollinators are generally steeper than those for honeybees [46,47,48]. The increasingly erratic weather patterns reported by beekeepers in this region are also likely to be having severe impacts on the wild pollinator community, as weather patterns are known to be one of the most important factors influencing wild insect populations [10]. In regions of the world where honeybees have experienced declines from Colony Collapse Disorder (generally linked to honeybee-specific diseases), wild pollinators have been shown to provide effective insurance against honeybee losses [49]. However, in our study region, where the reported drivers of honeybee decline are stressors which also impact wild pollinators (loss of flowers, changing weather patterns and pesticides), it is unlikely that wild pollinators will be able to compensate for declines in the pollination services provided by honeybees. This increases the potential vulnerability of pollination services in this region and further emphasizes the importance of mitigating these declines.

### 4.4. Knowledge Gaps, Future Prospects and Management Advice

To mitigate further declines in *Apis cerana cerana* and wild pollinators in this region, farmers should be made aware of the value of these insects for their crop production, and encouraged to incorporate pollinator-friendly management practices into their farming. Pollinator-friendly management may include the increased provision of floral resources, reduced insecticide and herbicide use, and the maintenance of crop diversity and natural habitat areas on their farms. Floral resources can be enhanced by incorporating native flowering shrubs such as *Cotoneaster microphyllus*, *Prinsepia utilis*, and *Rosa sericea* into hedgerows and adding flower-rich field margins to farmland [23]. These should ideally be provided in close proximity to hives and crops so that bees do not have to travel far and run the risk of getting caught during bad weather. A previous study from Nepal shows that comb building in *A. cerana* colonies accelerated in the vicinity of mustard *Brassica* spp. and buckwheat *Fagopyrum esculentum*, two mass-flowering crop species grown in Jumla [22]. It is also important to ensure adequate nesting sites for both wild-nesting honeybee colonies and other wild pollinators; these may include old trees and logs, piles of sticks, and bare sloping earth banks. Due to the increasing occurrence of unpredictable weather, which is likely to become more common in the future under climate change, beekeepers may benefit from using well-insulated log hives or modified log top-bar hives which buffer colonies from extreme temperature fluctuations and are resistant to attack by pine martens. We discourage the use of all chemical pesticides, especially during the flowering season. Additionally, it is essential that beekeeping efforts in this region utilize the native *Apis cerana* species and avoid the import of *Apis mellifera*, which might exacerbate the decline in the native bees and pollinators through the spread of disease and competition for diminishing floral resources [21,50].

The household survey and our key informant interview indicated that beekeepers have limited knowledge about the importance of bees for crop production. Thus, an important priority in the region should be to increase public awareness of the importance of insect pollination through a widespread education and outreach program. Moveover, a training and extension program to promote the management of bee colonies in movable comb hives would be of great value and could build on past successes with the Jumla top-bar hive—an adapted form of the traditional log hive. We know that *Apis cerana cerana* has been negatively affected by the brood diseases (e.g., Thai sacbrood and European foulbrood) since the 1990s [51] and it is likely that more recent agrochemical use in Jumla is exacerbating this. However, the current prevalence of diseases and agroechemical use is unknown, and the contribution of these factors to bee decline is not clear, therefore an expert assessment of bee disease prevalence and the relative contribution of pesticide poisoning as well as bee disease in this region should be undertaken.

## 5. Conclusions

Our study indicates that the native and semi-domesticated honeybee *Apis cerana cerana* is declining in Western Nepal. The exact reasons for the decline are unknown, but the respondents of our survey point to climate change, unpredictable weather patterns and reduced floral food resources as the main drivers. Further studies should be conducted to clarify the reasons for the decline, as well as the magnitude and extent of it. Based on the perceptions of respondents, the decline may have already impacted crop production in the region, as well as reducing revenue from honey production. Moreover, it is likely that the reported declines in honeybee populations and honey yields are indicative of a wider decline in wild pollinators in the region, especially if the causes of the decline are connected to climate change and reduced floral resource availability. To mitigate further declines in the honeybees and wild pollinators, we strongly advocate for the promotion of beekeeping with the indigenous *Apis cerana cerana* and the use of flowering hedgerows and field margins in proximity to crop fields and beehives, aswell as the maintenance native habitat areas to enhance food supplies and nesting sites for pollinators. We also recommend the use of well-insulated log or top-bar log hives to buffer bees against extreme temperature fluctuations, which are expected to worsen with climate change.

## Figures and Tables

**Figure 1 insects-15-00281-f001:**
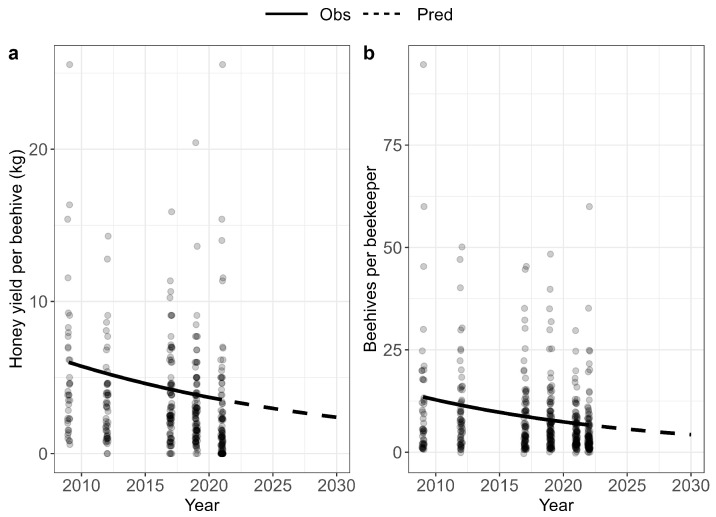
Change in (**a**) honey yield per hive (kg) for the period before 2012 to 2022 and (**b**) number of beehives per beekeeper for the period before 2012 to 2021 across all villages (Obs), with stippled prediction lines (Pred) until year 2030. The analyses are based on 116 beekeeper respondents.

**Figure 2 insects-15-00281-f002:**
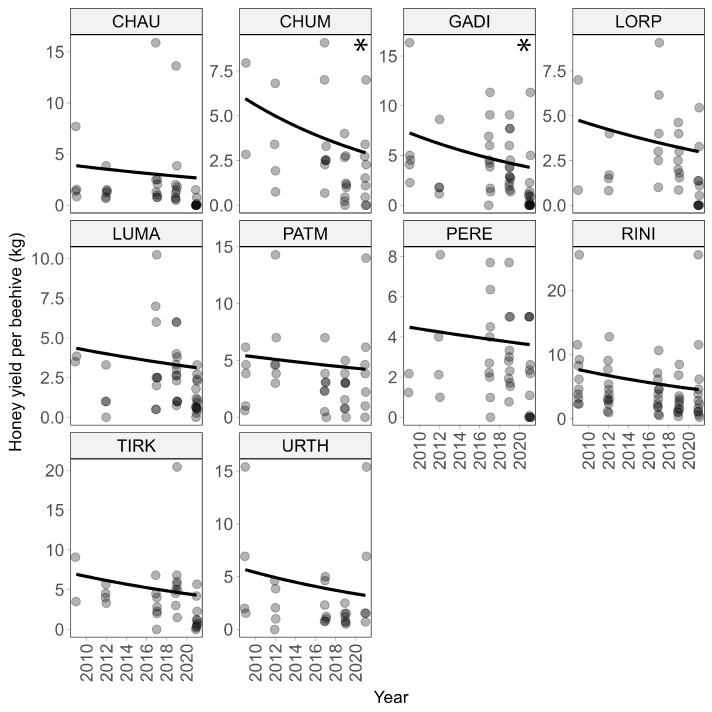
Changes in honey yield per beehive for the 10 villages (on average 11 respondents per village) for the period before 2012 to 2021. The decline in honey yield was consistent across villages but only statistically significant (*p* < 0.05) for two villages (Chuma and Gadigaun) marked with an asterisk in the panels upper right corner. Darker points in the panels are due to overlap in values.

**Figure 3 insects-15-00281-f003:**
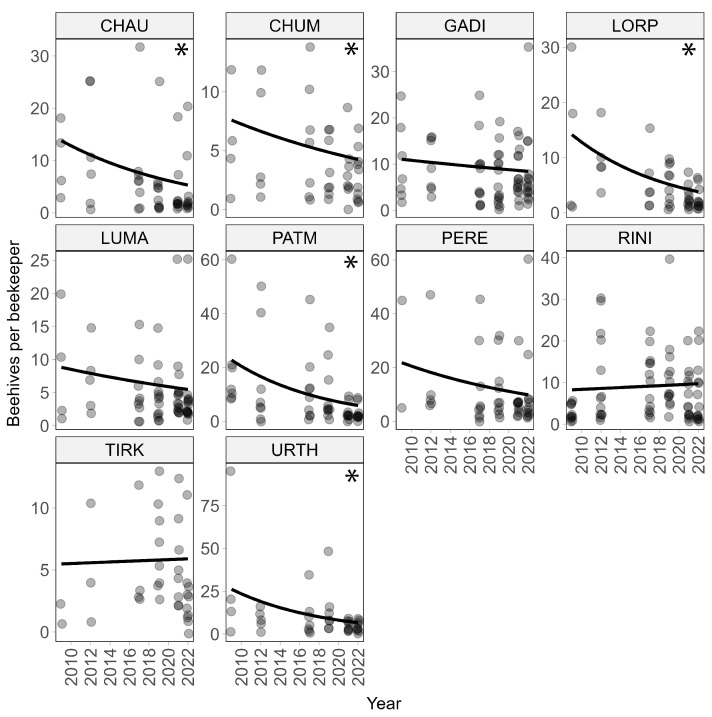
Changes in the number of beehives per beekeeper for the 10 villages (on average, 11 respondents per village). The decline in beehives per beekeepers was statistically significant (*p* < 0.05) for five villages (Chaura, Chuma, Lorpa, Patmara and Urthu) marked with an asterisk in the panels upper right corner. Darker points in the panels are due to overlaps in values.

**Figure 4 insects-15-00281-f004:**
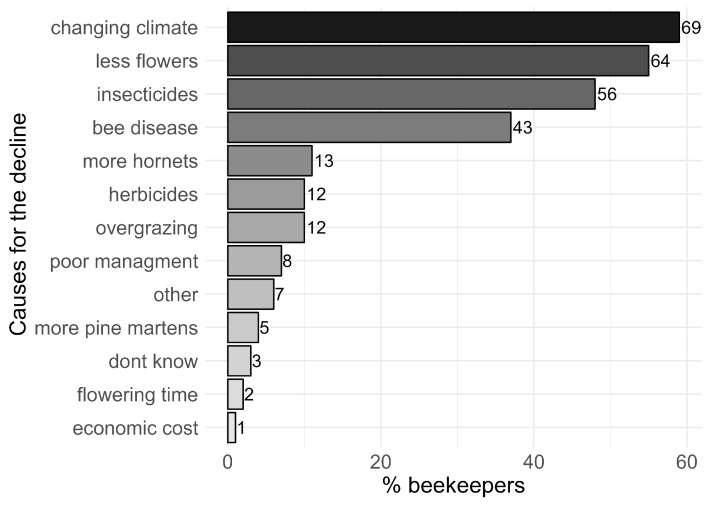
Ranking of the causes for the decline in honeybees according to beekeepers’ perceptions. Bar lengths are proportional to the percentage (*x*-axis) of beekeepers who listed a driver as a potential cause for the honeybee decline. The corresponding numbers of beekeepers are placed next to the bars. The top four causes for the decline perceived by the beekeepers are (1) climate change, (2) fewer flowers, (3) insecticides, and (4) bee disease.

**Figure 5 insects-15-00281-f005:**
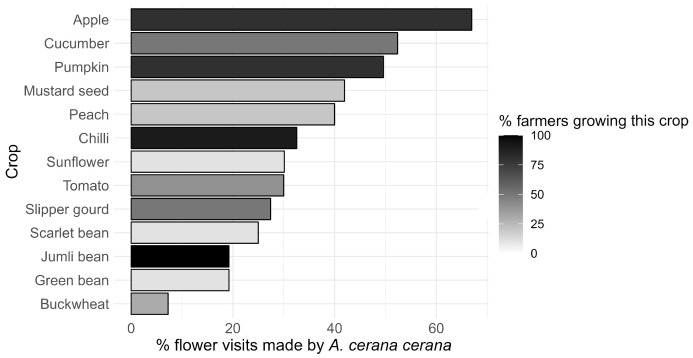
Percentage of visits from honeybees to the most important crop plants in the Jumla District. The color shading shows the percentage of farmers growing the crop.

**Table 1 insects-15-00281-t001:** Coefficients (β) and *p*-values for generalised linear models with a log link function relating honey yield or number of occupied beehives against year. Honey yield was modeled with an identity (Gaussian) link function while beehives were modeled with a log (Poisson) link function. One was added to the observed values to account for zeros.

	Honey Yield		Beehives	
**Term**	β	* **p** * **-Value**	β	* **p** * **-Value**
Intercept	90.11	<0.001	112.3	<0.001
Slope	−0.044	<0.001	−0.055	<0.001
McFadden’s pseudo-$R^{2}$	0.010		0.048	

## Data Availability

Data will be available from the NERC Environmental Information Data Centre (EIDC). Meanhwile, data and R scripts to conduct the analyses are downloadable from GitHub: github.com/skortsch/Honeybee _decline _Nepal via Honeybee decline in Nepal-v1.0.0, Zenodo: https://zenodo.org/doi/10.5281/zenodo.10973041, accessed on 10 April 2024.

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
