# Peer review of "Decline in Honeybees and Its Consequences for Beekeepers and Crop Pollination in Western Nepal"

_insects, 2024, doi:10.3390/insects15040281_

Round 1
Reviewer 1 Report
Comments and Suggestions for Authors
Manuscript presents results of questionnaire survey in Patarasi Rural Municipality, Jumla District of Mid-Western Nepal. Results are interesting in relation to decline of Apis cerana in Asia and similarity with problems in Europe and America. Study seems to be complete and describing complex of socio-economic and ecological aspects. The manuscript is well written and comprehensible. Results of this survey are valuable in a global context.
INTRODUCTION - aim of study
Aim of the study must be formulated more properly in the last paragraph of Introduction. I am sorry, it seems to be shallowly and awkwardly specified. Chiefly, the methological details, e.g. the questions to beekeepers, must be removed from this paragraph.
LINE 97. Apis mellifera himalaya is nomen nudum (Engel, 1999).
· Apis indica skorikovi Maa 1944: 4. Nomen nudum.
· Apis cerana himalaya Smith 1991b: 154. Nomen nudum.
· Apis cerana skorikovi Engel 1999:180. Nomen novum pro Apis indica skorikovi Maa 1944.
· Apis cerana himalayana Hepburn, Smith, Radloff et Otis 2001: 6. Nomen nudum.
Authors must use valid name i.e. A. m. skorikovi Engel 1999 which was installed as a nomen novum for this montanious subspecies of Apis cerana!
LINE 223
Could authors specify reasons for destruction of the deciduous flowering shrub Prinsepia utilis, please? This information is without some explanation incomplete.
BUCKWHEAT - attractivity to honeybee
I suggest that authors should outline in discussion why percentage of visits from honeybees to buckwheat was so extremelly low in comparison with doi:10.1071/ea99008 or Racys J., Montviliene R. 2005: Effect of bees-pollinators in buckwheat (Fagopyrum esculentum Moench) crops. Journal of Apicultural Science 49(1):47-51.
LINE 271
Authors should specify number colonies kept by an informant beekeeper (a local expert) in order to get an idea of the size of his apiary. Is it possible at least about any interval within past years? His colonies are mentioned at line 272-273.
LINE 360
Is it possible to provide any explanations for surprising result that only 17% of respondents were aware of the value of honeybees or other pollinators to crop production? Why beekeepers are not aware of this key knowledge? It would be useful briefly describe any reasons in addition to information at lines 393-395 and 416-419.
HONEY YIELD
It would be useful to comment in the discussion likely reasons for the honey yield per hive and year over 15 kg in some years. Is it explanable by any specific conditions at these apiaries?
Author Response
Please see the pdf attachment with responses to reviewer 1

Reviewer 2 Report
Comments and Suggestions for Authors
This work is of great importance and relevance in relation to the decline in bee populations and in particular to the livelihood that these bee populations support in Nepal, both as crop pollinators and honey producers.
It is hard work that contains relevant information about the problem in question.
Authors should note the following corrections:
Line 163 and line 168: "Insect visitation surveys", the authors did not do surveys about the visiting insects, they carried out sampling
Fig 1, 2 and 3: The authors must complete how much data they contribute to the information presented in the figures. On the other hand, this information would be clearer if a table with summarized data is added.
Fig 4: The authors should clarify the information, in the text it says that 69% of those surveyed answered that climate change affects the decline of bees and in the figure they indicate that 69 is the number of beekeepers surveyed. What is the correct information?
Line 258-261: Are the authors commenting on another survey or is it the survey they conducted in this work? If it is a comment on another survey, this information should appear in the discussion, not in the results.
Author Response
Please see the pdf attachment with responses to reviewer 2

Reviewer 3 Report
Comments and Suggestions for Authors
I consider the article: Decline of honeybees and its consequences for beekeepers and crop pollination in western Nepal should be accepted.
This research confirms the relevance of beekeepers' records in understudied regions of the world, in this way key information is obtained to take future actions.
I highlight the importance of having this type of studies to understand the impact of climate change and possible agricultural practices to the detriment of native bee populations. I do not doubt that these results will serve to promote environmentally friendly practices, select bee management strategies that allow the protection and multiplication of A. cerana populations.
Author Response
Please see the pdf attachment with responses to reviewer 3
